# Non-Coding RNAs: Regulators of Stress, Ageing, and Developmental Decisions in Yeast?

**DOI:** 10.3390/cells13070599

**Published:** 2024-03-29

**Authors:** Michal Čáp, Zdena Palková

**Affiliations:** Department of Genetics and Microbiology, Faculty of Science, Charles University, BIOCEV, 128 00 Prague, Czech Republic

**Keywords:** yeast, tRNA, lncRNA, RNA modifications, epitranscriptome

## Abstract

Cells must change their properties in order to adapt to a constantly changing environment. Most of the cellular sensing and regulatory mechanisms described so far are based on proteins that serve as sensors, signal transducers, and effectors of signalling pathways, resulting in altered cell physiology. In recent years, however, remarkable examples of the critical role of non-coding RNAs in some of these regulatory pathways have been described in various organisms. In this review, we focus on all classes of non-coding RNAs that play regulatory roles during stress response, starvation, and ageing in different yeast species as well as in structured yeast populations. Such regulation can occur, for example, by modulating the amount and functional state of tRNAs, rRNAs, or snRNAs that are directly involved in the processes of translation and splicing. In addition, long non-coding RNAs and microRNA-like molecules are bona fide regulators of the expression of their target genes. Non-coding RNAs thus represent an additional level of cellular regulation that is gradually being uncovered.

## 1. Introduction

In recent years, a growing number of studies have uncovered new roles for non-coding RNAs in cellular regulatory processes in diverse organisms across all kingdoms of life, from bacteria to mammals (reviewed in [1,2,3,4,5,6]). These newly identified processes involve not only groups of non-coding RNAs that are assumed to have mainly regulatory functions, such as microRNA (miRNA) and miRNA-like molecules and long non-coding RNA (lncRNA), but also RNA molecules that have been known for decades to have primary functions in essential cellular processes, such as translation and mRNA splicing. Recent discoveries regarding the role of known RNA classes, including, mainly, transfer RNA (tRNA) and, to a lesser extent, ribosomal RNA (rRNA) or small nuclear RNA (snRNA), shed new light on these molecules as players at an additional level of complexity in the mechanisms of cellular signalling and regulation. In particular, the emerging field of epitranscriptomics, which focuses on post-transcriptional nucleotide modifications of various RNAs, is currently growing rapidly, thanks to advanced methods for detecting modified bases (reviewed in [7]) and the current technology of direct sequencing using nanopore [7,8]. Interestingly, the regulatory role of RNAs has been uncovered in many cases under circumstances that require a complex cellular response, such as cellular differentiation or response to environmental challenges in the form of nutrient deprivation or various stress insults [1,9,10,11,12].

In this review, we focus on the role of different types of non-coding RNAs in yeast (Table 1) in regulatory processes and signalling pathways in response to stress conditions. Stress resistance is closely related to cellular longevity in yeast as well as in other organisms, including mammals [13,14,15,16]. In addition, cell differentiation and various lifestyle changes, such as the transition from yeast form to hyphae/pseudohyphae, are important determinants for the formation of multicellular structures in which cells diversify into different types and are often more resilient to stress and protected from hostile environments [17,18,19]. Here, we present examples of such regulations by non-coding RNAs in different yeast species. Because most research focuses on *Saccharomyces cerevisiae*, we describe here the findings discovered in this organism, unless otherwise stated.

## 2. Transfer RNA—More Than an Adapter

Measured by the number of molecules per cell, transfer RNA is the most common type of RNA. Each yeast cell, for example, contain several million of these molecules [20]. tRNA molecules have a typical secondary structure, often referred to as a “cloverleaf”. It consists of a double-helical acceptor stem and three main arms, each consisting of a short double-helical part that ends with a loop. These are the dihydrouridine arm (D arm, closer to the 5′ end), the anticodon arm (in the middle, containing the anticodon triplet), and the thymidine arm (T or TΨC arm, closer to the 3′end). The acceptor stem ends with the typical 3′-end sequence CCA, which provides a hydroxyl group (either 2′ OH or 3′ OH) for the binding of aminoacyl by aminoacyl-tRNA synthetases.

Transfer RNA plays a central role in the process of translation by decoding codons in messenger RNA (mRNA) into the amino acid sequence of synthesised proteins. Since the discovery of its function, tRNA has been considered only a ubiquitous mechanistic adaptor, but its cellular role may be much more complex.

New discoveries in the field of tRNA biology in recent decades have led to a model in which the cell can modify the translation profile by altering the amount or properties of tRNAs (reviewed in [21,22,23,24,25]) (Figure 1). According to this model, because the rate of translation depends on the concentration of cognate aminoacyl-tRNAs, differences in the amount of individual tRNAs lead to an alteration in the decoding capacity (i.e., the ability to translate mRNA with a certain codon composition) of the cellular tRNA pool and thus to faster translation of mRNAs whose codon composition matches the composition of the cellular tRNA pool. Consequently, changes in tRNA composition caused by selective transcription or selective degradation of individual tRNA species can completely alter the population of actively translated mRNAs and thus remodel the proteome even without changes in the mRNA transcriptome. The term tRNAome was introduced to refer to a set of all tRNAs present in the cell at a given time [26]. The effectiveness of the use of a tRNA molecule in the translation process is determined not only by the amount of a particular tRNA type (its synthesis and degradation) but also by its charge status, its subcellular localisation, and its post-transcriptional modifications. The latter is of particular importance as post-transcriptional modifications are important for many aspects of tRNA function, including the stability and efficiency of codon–anticodon interaction [27,28].

The analysis of tRNAome composition is often based on sequencing techniques, but these can be biased by the presence of modified nucleotides in the tRNA [29]. Hybridisation methods, on the other hand, are less sensitive to the presence of modified nucleotides. The estimation of tRNA abundance by these two methods correlates rather poorly, which calls into question the accuracy of the tRNAome sequencing approach [30]. In addition, Nagai et al. found a strong correlation between the abundance of individual tRNA and the number of copies of the corresponding gene in the genome using the hybridisation method, suggesting a lack of regulation of the abundance of individual tRNAs [30]. On the other hand, the study also found differences in the tRNAome between exponentially growing cells and cells in stationary phase. Thus, further research is needed to find out to what extent and by what mechanisms the yeast cells can change their tRNAome and whether the tRNAome changes lead to codon-dependent remodelling of translation in vivo.

The codon-composition-dependent translation rate can also have additional effects on protein folding and stability. It was shown that a translational pause caused by the presence of rare codons between individual domains of a protein provides sufficient time for proper folding of the freshly synthesised domain before the next domain is synthesised. In this case, replacing the rare codon with the optimal codon resulted in an increased misfolding rate [22,31]. On the other hand, it has been shown in mammalian cells that hydrophobic stretches in some proteins lead to protein aggregation when they are translated slowly [32]. Thus, some proteins can be stabilised and others destabilised by altering the overall rate of translation or the effectiveness of decoding individual codons by tRNA availability and functional state (i.e., modification and aminoacylation).

In addition, the rate of translation can be sensed by the cell and can lead to the destabilisation or repression of slowly translated mRNAs [33], thus further contributing to selective codon-dependent protein production. The correlation between optimal codon content and mRNA half-life has been observed in various organisms from yeast to mammals [33,34,35,36,37,38] and also appears to be influenced by the UTRs (untranslated regions) and the length of a particular mRNA. For example, this correlation was not observed for mRNAs coding for short peptides [39], which are preferentially translated by monosomes, whereas longer proteins are translated by polysomes [39,40].

The existence of such regulations could explain why different genes in the same organism have markedly different codon compositions. Moreover, codon-composition-mediated translational remodelling can be one of the factors contributing to the weak correlation between the mRNA transcriptome and proteome observed in many cases [41,42,43]. Evidence of the possible regulatory roles of tRNA in the context of stress, starvation, and ageing in yeasts is described in the following sections.

### 2.1. Abundance of tRNA—Synthesis and Degradation

The genome of *S. cerevisiae* contains genes for 42 different tRNA species, most of which are present in multiple copies at 275 predicted loci [44]. Transcription of tRNA genes by RNA polymerase III (RNA Pol III) is downregulated by starvation and stress, leading to an overall decrease in tRNA levels and a slowdown in translation. Under these conditions, the nutrient-sensing kinases TORC1 (Target of Rapamycin Complex 1), PKA (Protein Kinase A), and Sch9 are inactivated, leading to hypophosphorylation and subsequent nuclear localisation of the RNA Pol III repressor Maf1 and to activation of two inhibitory kinases for RNA Pol III—Mck1 and Kns1 [45]. Moreover, starvation blocks nuclear export of tRNA and even induces their relocalisation into the nucleus [46].

Although transcription by RNA Pol III has long been considered non-discriminatory and uniform for all RNA Pol III promoters, recent data show that the situation is more complex. For example, while stress conditions reduce the expression of most tRNA loci via Maf1, some tRNA genes seem to be more resistant or even insensitive to Maf1-mediated repression, leading to a modification of the tRNAome [47,48]. The mechanism of this differential regulation is unclear, as all tRNA genes use the same simple transcription factor machinery that recognises simple promoter boxes within the transcribed region. One possible mechanism relies on the chromatin context of individual tRNA genes. Recently, it was discovered that the chromatin remodelling complex FACT (Facilitates Chromatin Transcription) is dynamically associated with tRNA genes in a stress-dependent manner [49]. Another study in mammalian cells suggests that transcriptional interference by RNA Pol II activity at promoters adjacent to tRNA genes is an essential player in the repression of tRNA genes [50]. In contrast to this model, a study in *Schizosaccharomyces pombe* showed that chromatin remodelling facilitated by RNA Pol II transcription induces transcription by RNA Pol III from neighbouring tRNA genes [51]. Thus, global changes in RNA Pol II transcription induced by stress and starvation can lead to changes in the tRNAome. Alternatively, specific DNA-binding proteins can also regulate the binding of the general RNA Pol III transcription factors TFIIIB and TFIIIC to individual tRNA genes similar to the mechanism described in mammalian cells [52]. Indeed, the regulator of yeast RNA Pol III assembly Fpt1 binds to promoters of some tRNA genes and modulates their transcription [53].

The role of tRNA abundance as a translational regulator under stress conditions has been supported by several studies in yeast. First, a different abundance of different tRNAs was observed under different stress conditions [54,55]. A comparison of the tRNA transcriptome with the translatome (i.e., the set of all mRNAs that are translated in the cell in a given time) under conditions of oxidative stress revealed that the amounts of individual tRNA species are positively correlated with the cognate codon content of the actively translated mRNAs. Moreover, a reporter protein that was codon-optimised for tRNA composition under stress conditions was translated more effectively upon hydrogen peroxide treatment than reporters with different codon composition [55]. Thus, the stress-induced changes in tRNA abundances are sufficient to alter the translational efficiency of the respective codons and may consequently be reflected in the global proteome.

Another example comes from a study on the regulation of the major B-type cyclin Cdc13, which drives the cell cycle and mitotic entry in fission yeast [24,56]. The gene *cdc13* contains the non-optimal glycine codons GGA and GGG, both of which are decoded by tRNA^Gly^_UCC_. Overexpression of tRNA^Gly^_UCC_ or replacement of the non-optimal codons with a preferred glycine codon resulted in a cell cycle defect and slow growth due to Cdc13 aggregation. Interestingly, the abundance of tRNA^Gly^_UCC_ increases under oxidative stress [24]. A possible model suggests that stress-induced tRNA production leads to cyclin inactivation due to an altered translation rate and thus to delayed cell cycle progression.

The amount of individual tRNAs at any given time is determined by the rate of their transcription and processing on one side and the rate of degradation on the other side, and it can change dynamically in response to cellular needs. Bulk tRNA degradation occurs under stress and starvation conditions by RNAse Rny1 [57,58]. However, there are indications that tRNA degradation can be selective towards a subset of cellular tRNA. Genome-wide mapping of ribosomal occupancy revealed translational pauses at specific codons under conditions of oxidative stress. These pauses did not occur in the strain lacking Rny1, suggesting that Rny1-mediated tRNA cleavage, which may be specific for some tRNA species, is able to achieve the translational changes in vivo [59].

Cleavage of tRNAs by endonucleases to form tRNA fragments (tRFs) occurs under oxidative stress and starvation and is conserved from yeast to plants and mammals [12,60,61,62]. The cleavage can occur in one of the loop regions of the tRNA, resulting in different tRFs. Cleavage in the TΨC-loop and the D-loop produces 3′ tRFs and 5′ tRFs, respectively, whereas cleavage in the anticodon loop produces tRNA halves [63]. The cleavage not only reduces tRNA abundance, which affects the translation rate, but the resulting tRFs also have additional functions. In mammals, tRFs are now well-established signalling molecules involved primarily in the regulation of gene expression through a mechanism of RNA interference (reviewed in [63]), but the function of tRFs in yeast is much more elusive. Genome-wide sequencing of stable, non-coding transcripts has identified many tRFs that are produced in the cell under stress conditions. The repertoire of these fragments differed significantly under different stress conditions, suggesting regulated selective tRNA fragmentation under different environmental conditions [64]. Because components of the RNA interference pathway are not present in *S. cerevisiae*, the function of tRFs there must be based on a different mechanism. It has been reported that many tRFs bind to the small ribosomal subunit and to the ribosome-associated aminoacyl-tRNA synthetases, thereby inhibiting translation and aminoacylation, respectively [65,66,67]. A similar composition of tRFs was found in *S. pombe* and two *Cryptococcus* species, *C. neoformans* and *C. gattii*, indicating evolutionary conservation between distant yeasts [68,69]. Furthermore, mitochondrial tRNA fragmentation is induced by heat stress and the stationary phase in *S. pombe*, establishing a link between environmental stress and respiratory capacity controlled by mitochondrial proteosynthetic capabilities [70].

### 2.2. Charge Status of tRNAs

One aspect of the involvement of tRNAs in cellular signalling has long been known and is related to the general amino acid control pathway (GAAC), which is mediated by the protein kinase Gcn2 and the transcription factor Gcn4. Uncharged tRNAs that accumulate under nitrogen starvation activate the GAAC pathway, leading to the production of the transcriptional master activator Gcn4, whose induction leads to the activation of amino acid biosynthetic genes, resulting in increased tRNA charging [71]. The presence of uncharged tRNAs is recognised by the histydyl-tRNA-synthetase-like domain of Gcn2, an evolutionarily conserved eIF2α kinase, leading to its activation. Gcn2 in turn phosphorylates the translation initiation factor eIF2, resulting in a reduced translation initiation rate. This decreases global protein synthesis, but, on the other hand, increases the synthesis of specific proteins, such as Gcn4, through a mechanism involving upstream ORFs (uORFs; reviewed in [72]). In addition, uncharged tRNAs directly inhibit the activity of TORC1 in *S. cerevisiae* in in vitro experiments [73]. In *S. pombe*, tRNA precursors, rather than uncharged mature tRNA, modulate the nutrient-sensing TORC1 pathway during the transition from vegetative growth to sexual reproduction [74].

In addition to the availability of amino acids, the charge state of tRNAs is also regulated by the activity of the individual aminoacyl-tRNA synthetases. The activities of many aminoacyl-tRNA synthetases are regulated by stress and nutrient availability either directly or through interaction with stress- or nutrient-responsive accessory proteins [75,76,77,78,79,80]. In this way, the cell can change the charge status of individual tRNA types. The level of uncharged tRNAs is an indicator that integrates metabolism and stress factors to adjust cellular responses via GAAC and TORC1 signalling pathways. In addition, the charge state of individual tRNA species governs their availability for translation and may therefore contribute to translatome remodelling via a mechanism described above (Figure 1).

### 2.3. Modifications of tRNA

Each type of tRNA is modified at different positions, with an average of 11 modifications per tRNA molecule in yeast. These modifications occur in all parts of the tRNA molecule. Some of them are conserved among all tRNA species in a given organism and also across evolutionary distant genera, while others are specific to a particular tRNA or a particular species [81]. Examples of the former are dihydrouridine modifications within the D loop and formation of ribothimidine (T) and pseudouridine (Ψ) within the TΨC loop. Modifications in the anticodon, particularly at wobble position 34, are among the most frequent and important for tRNA function, as they ensure correct codon–anticodon pairing and dual-codon recognition and thus influence the translation rate and fidelity [27,82]. Some modifications are simple reactions (e.g., methylations) catalysed by a single enzyme, while others require multistep biosynthetic pathways carried out by large enzyme complexes. An example of such a complex dual modification is the addition of a methoxycarbonylmethyl group to carbon 5 and a thiolation of carbon 2 in uracil at position 34 (mcm^5^s^2^U34).

The cells have a large enzymatic machinery for carrying out tRNA modifications, which, in *S. cerevisiae,* consists of at least 73 proteins (representing more than 1% of the genome) [83]. Therefore, the cells can modify the tRNA epitranscriptome by altering the amount and activity of tRNA-modifying enzymes. Hypomodified tRNAs are less effectively used for translation, are more likely to be degraded, and are less aminoacylated or even misacylated [84,85,86]. Consequently, changes in the tRNA epitranscriptome can shift the decoding capacity of the tRNA pool, leading to reprogramming of translation due to different translation rates in different mRNAs, an effect comparable to changes in the concentration of different tRNAs.

This hypothesis is supported by the transcriptome-wide data showing that different stress conditions induce specific changes in tRNA modification patterns [87,88] and by the findings that reduced modification of some tRNAs leads to reduced translation rates at cognate codons [89]. Quantification of modified nucleotides in tRNAs under different stress conditions revealed that the extent of methylation of certain nucleotides is increased under oxidative stress but not under other stress conditions, and that these modifications are required for the proper response to oxidative stress [87]. The subsequent study focused on tRNA^Leu^_CAA_, which is hypermethylated at wobble cytosine 34 under oxidative stress. Reporter protein and proteomic analyses showed that this oxidative-stress-induced methylation indeed increased the efficiency of translation of UUG containing mRNAs [90]. Similarly, increased methylation of cytosine 32 in tRNA^Thr^_GGU_ induced by the treatment of cells with the alkylating agent methyl methanesulfonate led to a concomitant increase in the amount of proteins enriched in cognate codons [89]. 

Further support comes from the findings that some tRNA-modifying enzymes are influenced by stress and nutrient availability. Urm1, which is involved in the thiolation of the wobble uracil at position 34 of tRNA^Lys^_UUU_, tRNA^Glu^_UUC,_ and tRNA^Gln^_UUG_ [91], is a highly unstable protein whose concentration is sensitive to translational perturbations caused by nutrient deprivation and stress [92]. In addition, the availability of the sulphur-containing amino acids cysteine and methionine in the cell is crucial for the function of Urm1 in the tRNA thiolation pathway [93]. Urm1 also plays a direct role in the removal of reactive oxygen species (ROS), as it is covalently bound to the peroxidase Ahp1 in a ubiquitin-like manner [94,95]. Thus, Urm1 may act as a sensor that integrates signals of nutrient availability and oxidative and other stresses and modulates tRNA thiolation in response to these signals, leading to altered decoding capacity and reprogramming of translation.

Reduced modification in some tRNAs can also lead to decreased translation fidelity, as the hypomodified tRNA can pair to non-cognate codons. For example, loss of the *N*^6^-threonylcarbamoyl modification at adenine 37 in tRNA^Lys^_UUU_ leads to increased misreading of the STOP codons UAA and UAG by the hypomodified tRNA^Lys^_UUU_ [96]. The resulting increased read-through at translation stop sites not only induces proteotoxic stress and the accumulation of dysfunctional proteins, but may also contribute to phenotypic diversity and the emergence of new phenotypes by expressing the sequences in the 3′ untranslated regions (3′UTR) that are silent under normal conditions [97,98]. 

Reduced tRNA modification can also lead to tRNA degradation. For example, hypomodified tRNA^Met^_i_ is degraded by a mechanism involving the nuclear TRAMP (Trf4/*A*ir2/Mtr4 Polyadenylation) complex and the exosome [99,100], demonstrating that the amount of a particular tRNA can be regulated by the activity of the modifying enzymes. The lack of modification of several other tRNA species leads to their degradation via the RTD (Rapid tRNA Degradation) pathway, in which exonucleases Xrn1 and Rat1 are involved [101]. The RTD pathway plays a similar role in *S. pombe*, suggesting an evolutionary conservation of this regulation [102]. Moreover, hypomodified tRNAs are effectively withdrawn from translation by their retrograde transport to the nucleus [103]. Thus, regulated modifications not only alter the functional properties of individual tRNA species during translation but may also be involved in the targeted adjustment of the amount of a particular tRNA. 

To serve as an effective regulatory mechanism, tRNA modifications should not only be produced in a regulated manner but also be removed in response to cellular signals. The only known tRNA de-modifying enzymes are the mammalian demethylases ALKBH1 and ALKBH3 [104,105]. In yeasts, no specific enzyme is known to remove tRNA modifications. Nevertheless, stress factors can have a direct effect on tRNA molecules. For example, thiolation at uracil 34 (s^2^U34) can be directly removed through reaction with ROS [106].

Taken together, regulation of the expression of specific tRNAs, their processing, modification, localisation, degradation, and aminoacylation could lead to differential efficiency in the translation of mRNAs with different codon compositions. Thus, relatively subtle changes, such as shifts in the abundance of individual tRNA species or in the extent of their post-transcriptional modifications, which require only a relatively small amount of energy and material, could alter the pool of translated mRNAs and lead to global changes in the cellular proteome. Individual tRNA species can be viewed as hubs that take up and integrate inputs from different pathways. The resulting output in the form of tRNA abundance and functional state then alters the translational landscape, modulates signalling pathways, and regulates other cellular processes (Figure 2). What was originally seen as merely a molecular adapter that brings the amino acid to its cognate codon on the mRNA also has properties of a regulatory molecule with functions beyond translation.

## 3. Long Non-Coding RNAs

Long non-coding RNAs (lncRNAs) are defined as RNA molecules that are longer than 200 nucleotides and are not translated into proteins [107]. They are a heterogeneous group of RNAs that are present in all kingdoms of life, and they are created by different processes and have different biological functions that are executed by a variety of mechanisms. LncRNAs have only relatively recently been recognised as a functional class of RNAs, although some evidence of transcription from DNA regions outside of the known genes has been reported previously. Probably the first well-described example of a functional non-coding RNA is Xist RNA, which plays a role in the inactivation of an X chromosome in female mammals [108]. Since then, in particular with the onset of next-generation sequencing technologies, thousands of lncRNAs have been found, and for many of them the mechanism of their function has been elucidated [107]. However, we are still far from fully understanding their role in the cell. Although most examples and functional diversity have been documented in mammalian cells, hundreds to thousands of lncRNAs have also been described in various evolutionarily distant yeast species [109,110,111,112,113,114]. The function of the vast majority of them has not been characterised, and there are only a few examples where the exact molecular mechanism underlying the function has been uncovered. It seems that yeast cells rely more on regulation by lncRNAs during stress, starvation, and developmental decisions than under optimal growth conditions. Non-coding antisense transcripts are enriched in environmentally controlled genes, and major changes in lncRNAs have been detected during environmental stress and nutrient starvation [115,116,117,118,119]. The mechanisms of lncRNA function in yeast are usually in cis, with the expression of a particular lncRNA affecting the expression of a neighbouring or overlapping gene, but some examples of trans-acting lncRNAs have also been described [120,121]. A comprehensive overview of this topic is provided in recent reviews [107,119,122,123]. In addition to the regulatory functions of lncRNAs discussed below, some studies show that some of these lncRNAs contain short open reading frames (ORFs), and the binding of ribosomes to these lncRNAs suggests that they could be translated, although nothing is known about potential peptides produced [124,125]. Similarly to the discovery of peptides with biological functions encoded by short ORFs (<100 codons) in the yeast genome, which were previously thought to be non-coding (reviewed in [126]), lncRNAs could thus be another source of such short peptides.

Here, we describe cases in which lncRNAs are involved in the stress response and longevity of various yeasts and in the regulation of lifestyle changes in multicellular yeast populations.

One of the mechanistically best understood examples of the action of lncRNAs in transcriptional regulation is the process by which lncRNAs regulate the expression of the *FLO11* gene, which encodes the surface glycoprotein adhesin Flo11. Flo11 is associated with various developmental processes in *S. cerevisiae* and is required for cell adhesion to various surfaces and invasive growth [127,128,129,130]. The function of Flo11 is essential for various multicellular phenotypes of strains that form structured multicellular communities, such as various types of biofilms, flocs, and mats. These multicellular structures protect their cell inhabitants from various environmental stresses and starvation through metabolic adaptation, diversification, and metabolite exchange within and between different cell subpopulations [17,18,131,132]. The regulation of *FLO11* expression is perhaps one of the most complex ones described to date in *S. cerevisiae*, reflecting the importance of lifestyle changes promoted by Flo11 in yeast populations. In addition to a very large promoter region that integrates multiple environmental signals [133], *FLO11* is regulated by lncRNAs (Figure 3a). Bumgarner et al. [134] identified two intergenic lncRNAs, *ICR1* and *PWR1*, upstream of the *FLO11* transcription start site, which are expressed from the sense and antisense strands, respectively. *ICR1* is located upstream of *FLO11* and covers most of its 3 kb promoter. Its transcription interferes with the transcription of *FLO11* and is negatively regulated by the transcription of the second lncRNA *PWR1*, which is expressed from the opposite strand and partially overlaps *ICR1*. The expression of *PWR1* is regulated by the positive transcriptional regulator Flo8 and the repressor Sfl1. The two transcription factors Flo8 and Sfl1 are regulated in opposite manners by the nutrient-sensing PKA pathway, which promotes filamentation. Thus, when Sfl1 is active, *ICR1* RNA is produced and *FLO11* expression is repressed. When Flo8 is active, the expression of *PWR1* represses *ICR1*, which in turn enables *FLO11* expression [134]. Expression of a similar pair of lncRNAs has also been detected upstream of *FLO10*, which encodes a different flocculin involved in different type of multicellular behaviour [134].

Filamentation and cell–cell adhesion are also controlled by lncRNA in the human pathogen *Cryptococcus neoformans* [135]. Here, the central protein for cell–cell adhesion, hyphae and biofilm formation, and colony morphology is the adhesin Cfl1, which is under the control of the transcription factor Znf2. Znf2 is able to sense environmental signals, such as stress, and, interestingly, also the presence of extracellular Cfl1 produced by neighbouring cells, which serves as a signalling molecule to coordinate the expression of adhesins within the population [135,136]. The upstream non-coding transcript *RZE1* regulates *ZNF2* transcription via an unidentified mechanism [135].

Osmostress-activated protein kinase Hog1 orchestrates the cellular response to perturbations in osmotic conditions by activating a variety of transcription factors. Together with these transcription factors, Hog1 binds directly to the promoters of target genes to stimulate their transcription initiation and elongation [137]. An important part of the Hog1-mediated stress response is the inhibition of the cell cycle through activation of inhibitors of Cdc28, a cyclin-dependent kinase (CDK) that controls cell cycle progression in *S. cerevisiae*. Hog1 also controls the expression of *CDC28* through an antisense lncRNA that completely overlaps the *CDC28* gene (Figure 3b). The expression of lncRNA is positively correlated with the expression of *CDC28* via the mechanism of DNA-loop-mediated activation [138]. In this model, Hog1 is initially active at an antisense promoter downstream of *CDC28* and recruits the RSC chromatin remodelling complex, which induces the expression of *CDC28* antisense lncRNA. These transcription-mediated chromatin changes stimulate the formation of a DNA loop between the *CDC28* promoter and terminator, mediated by Ssu72, and the transfer of Hog1 to the *CDC28* promoter, leading to the production of Cdc28. Thus, Hog1 inhibits Cdc28 at the protein level but induces the expression of *CDC28*. These seemingly opposing processes lead to cell cycle arrest but prepare the cell for re-entry into the cell cycle once it adapts to stress or the conditions become favourable again [137].

In *S. cerevisiae*, more than 1400 lncRNA transcripts have been identified by both microarray-based and sequencing technologies [116,139]. They can be categorised into two basic groups according to their stability. Stable unannotated transcripts (SUTs) and cryptic unstable transcripts (CUTs) are predominantly produced by bidirectional transcription from promoters of some genes and generally act in cis to regulate the expression of neighbouring genes, although at least four SUTs have been reported to act in trans [140]. Functional profiling of deletion strain collection revealed that many of the lncRNAs (including the four SUTs mentioned above) are required for proper fitness and survival under different environmental conditions [120].

The genomic study in *S. pombe* revealed changes in the expression of many lncRNAs under osmotic stress conditions [118]. More than two hundred antisense transcripts were identified, with enrichment in those overlapping genes involved in the general stress response and the specific response to osmotic stress. The respective protein levels were anticorrelated with the lncRNA/mRNA levels, suggesting that in most cases, transcription of antisense RNA reduces the expression of a neighbouring or overlapping gene (i.e., they are cis-acting) [118]. However, some of the identified lncRNAs act in trans. Deletion of *SPNCRNA.1164* resulted in resistance to oxidative stress, likely due to increased expression of distant genes *atf1*, *atf21*, and *atf31*, which encode key transcription factors downstream of the stress-activated MAP kinase Sty1 pathway, which promotes cellular responses to environmental stress and starvation. *SPNCRNA.1164* is one of few well-documented and confirmed examples of trans-acting lncRNA in yeast [118].

Atf1 also regulates the lncRNA-mediated expression of a key enzyme of carbon metabolism, fructose-1,6-bisphosphatase Fbp1, through carbon starvation. In this case, Atf1 and other transcription factors control the expression of sense lncRNAs with variable lengths (called mlonRNAs) upstream of *fbp1* (Figure 3c) [141]. These transcription events alter the chromatin state of the *fbp1* promoter region and make it more accessible to transcription initiation factors, leading to transcription of *fbp1* [141,142]. The involvement of lncRNAs in the stress response in *S. pombe* was further deciphered through high-throughput functional profiling of a library of strains in which one of ca. 150 intergenic lncRNAs (termed lincRNA) was deleted or overexpressed under a variety of different stress and nutrient conditions. Most lincRNA alterations (60% for the deletion library and 90% for the overexpression library) exhibited a phenotype under at least one condition [143].

LncRNA is also involved in the regulation cellular senescence in *S. cerevisiae*. One of the hallmarks of replicative ageing in yeast is the formation of extrachromosomal rDNA circles (ERC) (reviewed in [144]). The genetic stability of rDNA loci, which controls ERC formation, is regulated by lncRNAs transcribed by RNA polymerase II from the intergenic spacer region in rDNA (Figure 3d). Bidirectional transcription is initiated at the promoter called E-pro [145] and is positively regulated by the transcription factor Spt4 [146]. The lncRNA transcription prevents cohesin binding to rDNA and promotes genomic instability, leading to recombinational formation of ERC and accelerated replicative ageing. Silencing of ribosomal DNA mediated by the histone deacetylase Sir2 blocks lncRNA transcription, increases rDNA stability, and leads to an extension of the replicative lifespan [146].

A large number of lncRNAs has also been identified in differentiated populations of yeast colonies. Traven at al. [147] discovered several lncRNAs that are differentially expressed in different colony cell subpopulations. In other studies, several hundreds of long non-coding transcripts were discovered to be differentially expressed in cells in the upper (U cells) and lower (L cells) regions of an aged colony or in the so-called aerial or root parts of a structured biofilm colony [119,148,149,150]. These were mostly antisense transcripts that were either coregulated or antiregulated with their respective coding mRNA. The lncRNA/gene pairs covered many cellular functions, from metabolism and signal transduction to regulation of the cell cycle and sporulation [119]. Despite the apparent differences in lncRNA expression, no clear conclusion can be drawn from these data, in part due to the fragmented nature of the current understanding of lncRNA’s mode of action. 

Inactivation of the lncRNA called DINOR in *Candida auris* resulted in DNA damage, filamentation and wrinkled appearance of colonies, decreased virulence, and sensitivity to various drugs and stress factors [151]. The expression of DINOR was induced by DNA damaging agents and various drugs and stress factors, suggesting a function of DINOR in multiple stress response pathways. A study of genetic interactions revealed a link to the TORC1 pathway, but the exact mechanism of DINOR activity remains unclear. Long ncRNAs may also play a role in *Candida* spp. virulence, as a bioinformatic meta-analysis of RNA-seq datasets found many non-coding RNAs that alter their expression [111]. A similar analysis in *Cryptococcus neoformans* revealed a similar picture of massive lncRNA transcriptome reprogramming under conditions simulating host infection and elevated temperature and increased levels of oxidants [113]. 

## 4. Other Types of RNA

### 4.1. Ribosomal RNAs

Experiments in *Escherichia coli* have shown that certain oxidation products of rRNA have an impact on the ribosomal translation rate. While oxidation of some bases decreased ribosomal performance, oxidation of other bases had no effect or even increased translation [152]. Furthermore, oxidative stress induced the formation of covalent rRNA–protein cross-links in the yeast ribosome, although the effects on ribosomal function remain unclear [153]. Sublethal doses of oxidants as well as genetic perturbations leading to oxidative stress resulted in cleavage of yeast 25S rRNA at a specific site, cutting off the so-called expansion segment 7 (ES7), an rRNA element on the surface of the ribosome [154]. Its absence does not affect the rate of translation, but it can alter the binding of ribosome-associated proteins because ES7 mediates the interaction of ribosomes with ribosome biogenesis factors, chaperones, and enzymes that modify the nascent polypeptides, such as acetyltransferase and methionine aminopeptidase [155,156,157]. The cleavage of ES7 is catalysed by iron-mediated ROS production through Fenton chemistry. Experimental evidence suggests that the Fe^2+^ ion involved in ROS production is bound to the ribosome and may direct ROS production to induce site-specific cleavage [158]. A hypothetical model suggests that ROS-mediated cleavage of ES7 alters ribosomal properties, possibly to adjust the production of specific proteins or remodel the entire proteome. The presence of an ROS-sensitive “fragile” site in 25S rRNA and oxidative modifications of rRNA may be part of the ROS-sensing mechanism that promotes changes in ribosomal properties in response to oxidative stress.

Various stress conditions and ageing also lead to the repression of rRNA expression and endonucleolytic cleavage of 25S rRNA at specific sites [88,159,160,161]. Interestingly, this response is mediated by an unknown signalling molecule or metabolite acting as a quorum-sensing molecule [162]. In various yeasts, including *C. albicans*, *S. cerevisiae,* and *S. pombe*, a significant increase in rRNA resistance to exonucleolytic 5′-3′ cleavage was observed upon cell entry into the stationary phase or treatment with the TORC1 inhibitor rapamycin [163]. The exonuclease resistance is conferred by the presence of the 5′ cap structure on rRNA molecules, and it is possible that capped rRNA is transcribed by RNA polymerase II. However, considering that ribosomal RNAs are transcribed as 35S precursors and post-transcriptionally spliced to 18S and 25S rRNAs, it is unlikely that canonical RNA Pol II-associated cotranscriptional capping is involved in the process [163,164]. The structure of the 5′cap and the mechanism of its formation are unknown. The capped rRNA may form functional ribosomes, leading to the attractive hypothesis that under starvation or stress conditions, a specific ribosome subtype with specific properties can be formed, thus enabling a specific mode of translation. 

Ribosomal RNA undergoes extensive modifications at approximately 100 nucleotides. The absence of some of these modifications may affect ribosomal functions, thus serving as a possible regulatory mechanism. However, in contrast to the results obtained with tRNA, direct sequencing of yeast rRNA under different environmental conditions revealed few differences in rRNA modification pattern, thus questioning the modification as a physiological regulator of ribosomal function [88,165]. In contrast, Liu et al. [166] described an interesting regulation between metabolism and rRNA modification. The adenines A1781 and A1782 of 18S rRNA are dimethylated under normal growth conditions. These adenines are only monomethylated under sulphur or methionine starvation conditions due to limited availability of the methyl group donor S-adenosyl-methionine. Methylation changes influence ribosomal properties and promote increased translation of mRNAs coding for proteins involved in sulphur metabolism, possibly in an attempt to restore intracellular sulphur levels [166]. The mechanism by which ribosomes selectively translate mRNAs related to sulphur metabolism is currently unknown. One possible mechanism involves the selection of specific mRNAs by RNA-binding proteins.

### 4.2. RNAs from Introns

RNA derived from excised introns plays an interesting role in growth regulation in *S. cerevisiae* [167,168]. An analysis of a collection of strains in which individual introns were deleted revealed that deletions of most introns, although not showing an obvious phenotype under nutrient-rich conditions, led to a reduced ability of cells to survive starvation [168]. Furthermore, most strains from this collection overgrew the wild-type strain when nutrients were resupplied. The intron deletion phenotype was independent of the function of the host gene but rather was linked to aberrant repression of ribosomal protein genes by a mechanism involving the nutrient-sensing kinases TORC1 and PKA [168]. While excised introns form lariat structures that are rapidly degraded in the nucleus under optimal growth conditions, the accumulation of unusual linearised forms of introns from 34 particular genes has been observed in cells from saturated dense liquid cultures and from cell lawns grown on solid medium [167]. Deletion of the introns of the five most prevailing genes resulted in a strain that exhibited reduced survival during starvation, which is consistent with the above-mentioned study by Parenteau et al. [168]. In addition, treatment with the TORC1 inhibitor rapamycin and the secretory stress-inducing chemicals tunicamycin and dithiothreitol also resulted in linear intron accumulation. Notably, this linear intron accumulation occurred under conditions of prolonged slow TORC1 inactivation, which differs substantially from the rapid TORC1 inactivation triggered by rapid nutrient depletion. The possible mechanism by which intron RNA regulates growth is by binding and regulating the function of the spliceosome [167,168]. Because introns are relatively rare in *S. cerevisiae* and are predominantly found in growth-related ribosomal protein genes, the inhibition of splicing represents a potential negative regulatory mechanism that affects ribosome production and thus cell growth [169]. A reduced amount of intronic RNA in strains with intron deletions thus leads to insufficient splicing repression and, consequently, to a higher expression of ribosomal protein genes, resulting in the strain’s inability to survive starvation. These two studies highlight intronic RNAs as regulators of cell growth that are integrated into nutrient-sensing signalling pathways.

### 4.3. Small Nuclear RNAs

The function of spliceosomes can be regulated by post-transcriptional modifications of spliceosomal small nuclear RNAs (snRNAs). For instance, U2 snRNA can undergo inducible pseudouridylation at positions 56 (Ψ56) and 93 (Ψ93) through the action of the pseudouridine synthase Pus7. Under normal growth conditions, these modifications are not present. Heat stress and starvation induce the formation of Ψ56 [170], whereas starvation in saturated cultures and rapamycin treatment induce the formation of Ψ93 [171]. Both modifications occur within a flexible stem region of U2 snRNA that is important for the catalytic function of the spliceosome. However, the exact biological consequences of these modifications are unclear. Pseudouridylation in U6 snRNA at position 28 by pseudouridine synthase Pus1 is specifically stimulated during growth on solid medium and under conditions that induce pseudohyphal growth. Cells lacking Ψ28 failed to produce pseudohyphae, and artificially increased pseudouridylation was sufficient to enhance pseudohyphae formation [172]. Hence, regulated pseudouridylation of snRNA can reprogram the spliceosome to alter gene expression during starvation and differentiation. Stress-induced and filamentation-related pseudouridylation can influence spliceosome function not only in terms of overall activity (likely inhibitory) but also in terms of recognising non-optimal and modified splice sites, thereby inducing alternative splicing or regulating the splicing of a specific subset of genes [170,172,173]. 

### 4.4. RNAs in Extracellular Vesicles

Many organisms across different kingdoms of life secrete extracellular vesicles (EVs) that contain various substances, including proteins, polysaccharides, lipids, and nucleic acids [174,175]. Secretion of EVs, which contain RNA along with other components, has been reported in many yeast species, including *S. cerevisiae* and the clinically important yeasts *Candida* spp. and *Cryptococcus* spp. [176]. EV secretion has been proposed as an important factor in the establishment of yeast’s protective mechanisms, such as modifications of the cell wall, production of the capsule and extracellular matrix, and even as a means of intercellular signalling. Especially in the latter case, RNA could play a pivotal role. Extracellular RNA and DNA have been detected in *C. albicans* biofilms [176,177], and the EVs’ secretion is likely a mechanism for transporting these nucleic acids out of cells. As shown for *C. albicans* infection, extracellular nucleic acids are recognised by the host immune system and trigger an ROS-mediated immune response [177].

A global analysis of the RNA content of EVs was performed in four yeast species: *P. brasiliensis*, *C. neoformans*, *C. albicans*, and *S. cerevisiae* [176]. In all species, EVs contained a mixture of different cellular small RNAs, mainly snoRNA, tRNA, snRNA and mRNA fragments, as well as miRNA-like molecules with homology to miRNAs from different organisms. The function of most extracellular RNAs, if any, is unknown. However, the presence of many small RNAs that can potentially function as miRNAs in different organisms raises the intriguing possibility that extracellular RNAs modulate expression in other cells through RNA interference, thus enabling communication within or between species. The role of EV-derived RNA in cell-to-cell communication has been described in the emerging pathogen *Cryptococcus gattii* [178]. EVs secreted by yeast cells during infection are phagocytosed by macrophages into phagosomes, where they induce the growth of non-virulent cryptococcal cells residing in the phagosome. Treatment of the vesicles with RNAse or protease diminished this effect, implying that RNA, along with specific proteins, plays a role in the long-distance transmission of a virulence signal. Nonetheless, the exact mechanism of signal transmission has yet to be elucidated [178].

Recently, a novel discovery revealed small-RNA-mediated cross-kingdom signalling that governs the interaction between *C. albicans* and the host’s immune system [179]. Upon infection with *C. albicans*, macrophages secrete EVs that contain miRNAs and other compounds. One of the miRNAs, hsa-miR-24-3p, enters *Candida* cells and induces hyphal growth, which is a crucial virulence factor, by lowering the concentration of the CDK inhibitor Sol1. The mechanism of action of hsa-miR-24-3p likely involves translational repression directed by the sequence homology between hsa-miR-24-3p and *SOL1* mRNA. Interestingly, one of the functions of hsa-miR-24-3p in human cells is RNAi-mediated repression of p27, a conserved CDK inhibitor homologous to *C. albicans SOL1* [180]. In addition, hsa-miR-24-3p secretion is stimulated by soluble β-glucans, the immunogenic components of fungal cell walls [179]. *Candida* cells can therefore recognise the presence of activated macrophages through this mechanism and exploit miRNA produced by the macrophages to regulate the expression of their own homologous gene, thereby increasing their virulence.

## 5. Conclusions

A growing body of evidence suggests that non-coding RNAs of various types may represent an additional mechanism by which cells alter their physiology in response to external stimuli. In addition to the traditionally studied protein-based regulators and the increasing interest in the regulatory roles of small metabolites (metabolic intermediates), research into the regulatory roles of ncRNAs is thus becoming another rapidly developing area of research. In the frequently identified examples of regulation in which ncRNAs are involved, changes in the external living conditions of the cell/organism are reflected in the cell physiology. It can therefore be assumed (although direct evidence is often lacking) that these regulations contribute to various processes, such as rapid cellular adaptations, escape from stress and other harmful conditions, or changes in the stationary phase where other mechanisms might be switched off. Several genome-wide screening studies have already revealed significant differences in the representation of ncRNA in differentiated cells of structured multicellular populations, such as colonies and biofilms. Their functions are largely unknown, but specific cell subpopulations within these structures influence and respond in a coordinated manner to developmental changes that are often associated with nutrient deprivation and an increase in stress factors. A future challenge is to determine whether non-coding RNAs could play an important role in the development and differentiation of these structures.

## Figures and Tables

**Figure 1 cells-13-00599-f001:**
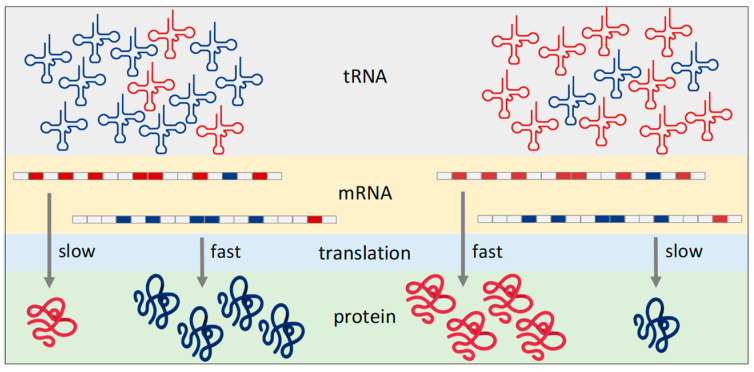
The model of regulation of translation rate by the availability of cognate tRNAs. The diagram illustrates the scenario in which a particular amino acid is encoded by two different codons (red and blue boxes) that are decoded by two different tRNAs (red and blue, respectively). In the situation shown on the left side of the diagram, the composition of the tRNAs favours the translation of mRNAs that are enriched in blue codons. The translation of mRNAs enriched in red codons is slow due to the relatively low concentration of cognate tRNA. Conversely, the concentration of the two tRNAs is reversed in the situation shown in the right part of the image. Consequently, mRNAs enriched in red codons are translated more effectively compared to mRNAs with blue codons.

**Figure 2 cells-13-00599-f002:**
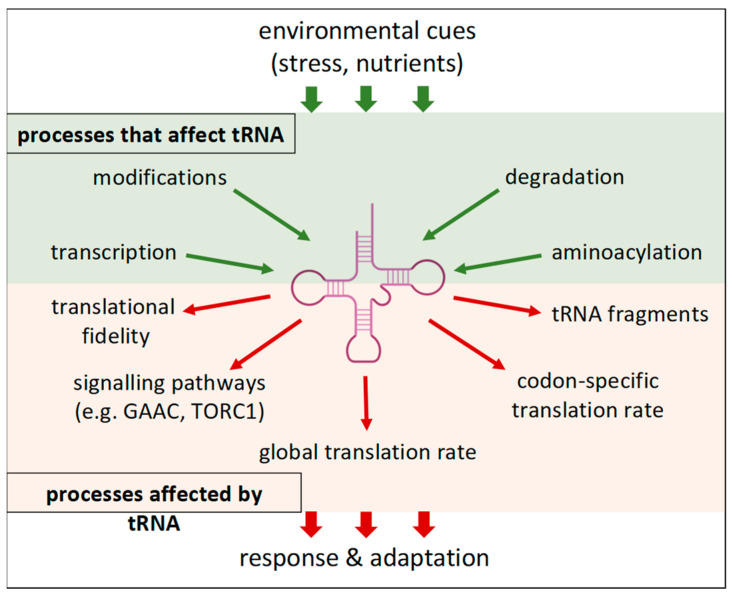
The role of tRNA in the regulation of cellular processes. Environmental stimuli, such as stress factors and nutrient availability, are sensed by signalling pathways (PKA, TORC1, and others) that regulate a group of effectors (among other targets), leading to changes in tRNA abundance, modifications, and other properties (green area, green arrows). These changes are reflected in altered properties of the translational apparatus and other outcomes that lead to overall changes in the cellular proteome (red area, red arrows), allowing the cell to adapt and respond appropriately to environmental conditions. See the text for details. GAAC, General Amino Acid Control; TORC1, Target of Rapamycin Complex 1; PKA, Protein Kinase A.

**Figure 3 cells-13-00599-f003:**
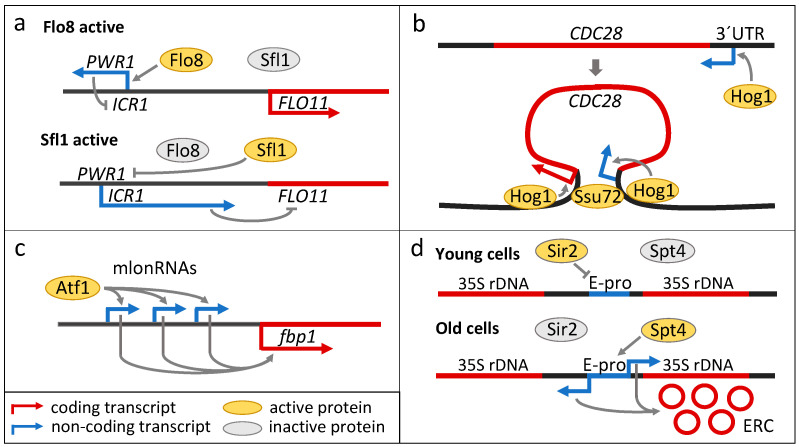
Examples of the regulatory role of lncRNAs in yeast developmental changes and stress. (**a**) Regulation of *FLO11* expression by the lncRNAs *PWR1* and *ICR1*, which are regulated by the transcription factors Sfl1 and Flo8. When the signalling pathways that induce cell filamentation are inactive, Sfl1 is active and Flo8 is inactive, leading to the expression of *ICR1*, which represses *FLO11*. Activation of signalling pathways that regulate filamentation leads to activation of Flo8 and inactivation of Sfl1, which eventually leads to expression of *FLO11*. (**b**) Regulation of *CDC28* expression by a DNA loop induced by the expression of an overlapping antisense lncRNA controlled by the kinase Hog1. (**c**) Regulation of *fbp1* expression by upstream lncRNAs, called mlonRNAs in *S. pombe*. Their expression is under the control of the stress-responsive transcription factor Atf1. (**d**) Induction of genomic instability in the rDNA locus by lncRNA expressed from the E-pro promoter. The transcription factor Spt4 and the histone deacetylase Sir2 regulate the transcription from E-pro.

**Table 1 cells-13-00599-t001:** Types of RNA with potential regulatory functions in yeast, how they are regulated, and what functions they may have.

RNA Type	Regulation	Possible Function
tRNA	modifications	translation rate
	stability	codon-dependent translation
	aminoacylation	
tRF *	production by	inhibition of aa-tRNA synthetases
	tRNA cleavage	inhibition of ribosomes
lncRNA *	expression	regulation of transcription
snRNA *	modifications	regulation of splicing
intronic RNA	linearisation and stabilisation	regulation of splicing
rRNA	cleavage modifications 5‘capping	regulation of ribosomal functions
miRNA-like *	expression	regulation of expression
evRNA *	secretion	various/unknown

* tRFs—tRNA fragments, lncRNA—long non-coding RNA, snRNA—small nuclear RNA, miRNA-like—microRNA-like, evRNA—RNA in extracellular vesicles.

## Data Availability

Not applicable.

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
