# Peer review of "Non-Coding RNAs: Regulators of Stress, Ageing, and Developmental Decisions in Yeast?"

_cells, 2024, doi:10.3390/cells13070599_

Round 1

Reviewer 1 Report

Comments and Suggestions for Authors In “Non-coding RNAs: Regulators of stress, aging and developmental decisions in yeast?” Michal ÄŒáp and Zdena Palková describe and discuss the findings concerning diverse classes of non-coding RNAs that play regulatory roles during the stress response, starvation, and aging in yeast. This work is well written, the text is clear and concise, and there are a lot of aspects to this work that I like. Nevertheless, the authors should address the following concerns.

1.       I am concerned with the term “micro RNAs” in the Abstract. Consistent with the findings described in chapter 4.4, authors should instead use the term “miRNA-like molecules” for RNA produced and secreted by yeast or/and host-secreted miRNAs for infectious yeast strains. Please clarify this issue.

2.       P2 L49 “Transfer RNA is the most numerous type of RNA in the cell.” It's a little confusing statement; please clarify

3.       In the Introduction, the authors state that their work describes the findings discovered in Saccharomyces cerevisiae unless otherwise stated. However, there are some places where data from other organisms are presented/discussed without appropriate indications, e.g., P3 L87-88 ref 21, P4 L144-146, ref 37…. It should be carefully checked.

4.       I think chapters 2.1 and 2.2, maybe also 2.3, need a better graphical summary than that presented in Figure 3. It would be helpful to readers to add a scheme of the GAAC pathway in the new figure. 

5.       P:4 L:180, please change to “the general amino acid control (GAAC) pathway.”

6.       In chapter 3, the authors described several yeast lncRNA. Understandably, authors focus on cis-acting lncRNAs as per se trans-acting lncRNAs in yeast are rare. Nevertheless, they mentioned S. pombe SPNCRNA.1164 without stressing that this is an example of trans-acting yeast lncRNAs (in oxidative stress). I would also suggest mentioning other trans-acting lncRNAs, e.g., SUT lncRNAs from S. cerevisiae.

7.       P:10, L:434 “expansion segment 7 (ES7), a non-essential rRNA element”. Indeed, its absence does not affect the TR, but it plays the other functions; please correct.

8.       FACT (L 137), TRAMP (L 247); please expand the shortcut

Reviewer 2 Report

Comments and Suggestions for Authors

Cap and Palkova present an overview of noncoding mRNAs in yeast, with special focus on their role in stress, aging and developmental decisions. The review is well written, but requires a few extensions, and clarifications.

1)      Transfer RNA p. 2-3

The authors discuss changes in tRNA abundance in different conditions. An important aspect of reviews is to point out limitations of conclusions or techniques, which is particularly true for the quantification of tRNA abundance. Their quantification is difficult because they are relatively short, strongly structured and modified and the similarity in their sequences can result in cross-reactions. Nagai et al developed a new method, OTTER, to quantify tRNAs and they show that tRNA abundance strongly correlates with tRNA copy number in yeast (Fig. 3). Importantly, they show in Fig. 6 that some of the methods, in particular, sequencing based ones, do not or correlate only weakly with OTTER. This study should be mentioned so that the reader can objectively evaluate the studies employing various tRNA quantification methods.

https://pubmed.ncbi.nlm.nih.gov/33674420/

2)      tRNA p 3 codon optimality and its association with translation and mRNA stability

“In addition, the rate of translation can be sensed by the cell and leads to the destabilization or repression of slowly translated mRNAs”.

This is true for most but not all mRNAs. There are mRNAs that have short coding sequences, which fail to change their stability and translation in response to codon optimality (Rahaman et al). These mRNAs have coding sequences shorter than 200-400 nucleotides, which include important genes in cell fate decisions such as the mating factor genes (MFA1 and MFA2) (Fig. 2), the above statement should be corrected/extended.

https://pubmed.ncbi.nlm.nih.gov/37756413/

3)       Flocculation p8

“In addition to an excessive promoter region”. The precise meaning of “excessive” is unclear. Do the authors mean that it is unusually long?

The authors discuss the regulation of this promoter but do not mention that flocculation is subject to developmental decisions, which constitutes the main focus of this review.  See Fig, 1 in:

https://pubmed.ncbi.nlm.nih.gov/15016375/

Comments on the Quality of English Language

English is OK. The word "excessive" should be replaced as indicated in comment 3.

Reviewer 3 Report

Comments and Suggestions for Authors

In this manuscript, the authors enumerate numerous examples where various non-coding RNAs regulate processes such as cellular stress, aging, or cell differentiation in various yeast species. Overall, the review is detailed and well-structured. However, I suggest some changes that could help better understand certain aspects of the work.

-          It would be convenient to include a paragraph or a summary table in the introduction discussing the different types of non-coding RNAs described in yeast.

-          The model described in Figure 1 should be supported by bibliographic references. However, the paragraph describing the model (lines 55-64) lacks any references to support it.

-          In the example described in lines 110-117, the function of the cyclin Cdc13 in the cell cycle should be included.

-          Include a brief paragraph on the structure and possible modifications of tRNAs.

-          In line 158, the authors propose that tRNA degradation produces tRNA fragments. However, numerous studies suggest that tRNA fragments (tRFs) are not merely degradation products but have specific functions. Therefore, I suggest including a paragraph on the structure, types, and synthesis of tRFs, emphasizing their biological functions beyond being considered degradation products of tRNAs.

-          In Figure 3A, the model indicates that Flo8 inactivates Sfl1 or vice versa. However, this is not what the authors of the study suggest, who propose a competition between both proteins. Therefore, I consider it necessary to modify the figure to remove the arrows connecting Flo8 and Sfl1.

-          Line 381, what function does this signalling pathway have?

-          Line 436, specify which properties of ribosomes are altered.

-          References 25, 54, and 104 do not have the same format as the rest.

-          There are entire paragraphs with hardly any references, such as lines 21-30 and lines 50-64.

-          In line 46, Saccharomyces cerevisiae should be italicized.

Overall, the manuscript is well-written and details significant advances in the field of ncRNAs. With some modifications, this review could be a significant contribution to the field of ncRNA-mediated regulation.

Round 2

Reviewer 2 Report

Comments and Suggestions for Authors

The authors addressed two out of the three comments. In the revised version, they discuss the reliability of different methods used to measure tRNA abundance (Comment 1) and mention that Flo11 is involved in a cell differentiation (comment 2).

They fail to address, however, comment 2 claiming that “mRNA stability and translation rate is incidental to the main subject of this review and is mainly mentioned to show further possible consequences of changes in translation rate.”.

It is difficult to believe that this subject is incidental to the authors’ manuscript because one out of the three figures is dedicated to this subject (Fig. 1). Even more important is that the authors discuss translation and mRNA decay in the context of non-coding RNAs. It is well known that many non-coding RNAs have been reclassified recently into short ORFs.

https://pubmed.ncbi.nlm.nih.gov/29465163/

As mentioned in my earlier review, the mRNAs with short ORFs show a distinct  relation between codon optimality and translation or mRNA half-lives. Therefore, it is in fact mandatory to discuss this aspect of mRNAs when reviewed in the context of non-coding mRNAs because otherwise the information shown in Fig.1 is partially misleading since further non-coding mRNAs are expected to be reclassified into mRNA with short ORFs. If it is difficult for the authors to add an estimated 2-4 sentences to clarify this topic, they should delete Fig. 1 and the associated content.

Author Response

We apologize for the misunderstanding in our response - we did not include the requested information in the revised manuscript as the aim of the review is not to discuss various aspects of stability of coding mRNAs, which can be affected by a spectrum of different mechanisms (RNA-binding proteins, polyadenylation, UTRs, etc.), including differences in stability in small coding ORFs such as MFA1/MFA2 mentioned in the first review. Nevertheless, we have included a brief information on mRNA stability with references in the new revised manuscript (pp.3-4, lines 125-131).

In this second revision, the reviewer states “Even more important is that the authors discuss translation and mRNA decay in the context of non-coding RNAs. It is well known that many non-coding RNAs have been reclassified recently into short ORFs. https://pubmed.ncbi.nlm.nih.gov/29465163/”.  Yes, since the first annotation of the yeast genome, where the short ORFs were not considered coding, many of them have been reclassified as coding for small proteins/peptides with biological functions (reviewed in the article PMID29465163 mentioned by the reviewer). As far as we know, these short ORFs are not currently expected to be a category of non-coding RNAs and are certainly not typical lncRNAs. On the other hand, we agree that some of the lncRNAs containing short ORFs could produce peptides - as suggested by the documented binding of ribosomes (e.g. PMID:24931603), although no function has yet been identified. We have included this information in Chapter 3 (p.9, lines 384-391).

Round 3

Reviewer 2 Report

Comments and Suggestions for Authors

The authors addressed the remaining point.